# Liquid-Nitrogen Spray-Cooling Combined Effect of the Spray Height and Heat-Sink-Surface Diameter

## Yangzi She * and Yanlong Jiang

Institute of Astronautics, Nanjing University of Aeronautics and Astronautics, Nanjing 210016, China; jiang-yanlong@nuaa.edu.cn
* Correspondence: sheyangzi@nuaa.edu.cn; Tel.: +86-186-6224-8400

**Abstract:** An experimental bench was constructed for liquid nitrogen spray cooling in an open loop system, and the combined effect of the spray height, heat sink surface diameter, and spray mass flow on the heat transfer characteristics of liquid nitrogen spray cooling was studied by comparing 50 sets of experimental data. In general, the heat transfer performance decreased with an increase in the spray height and heat sink surface diameter, with the latter's influence being more dominant. In the experiment, when the heat sink surface diameter increased to 24 mm, the influence of the spray height on the heat transfer performance decreased. The maximum critical heat flux and maximum heat transfer coefficient were 191.90 W·cm$^{-2}$ and 16.67 W·cm$^{-2}$·K$^{-1}$.

**Keywords:** spray cooling; liquid nitrogen; spray height; surface diameter; combined effect

## 1. Introduction

Spray cooling is widely used for electronic equipment cooling and in high energy lasers, advanced radar systems, particle accelerators, and so on, owing to its small heat transfer temperature difference, absence of boiling hysteresis, and high temperature uniformity of the cooling surface [1]. During spray cooling, the spray nozzle generates a large number of high speed small droplets, which can fully cover and exchange heat with the heat sink; the heat exchange results in the droplets' phase being transformed. Consequently, spray cooling can remove high heat flux and maintain a uniform surface temperature [2,3].

The heat transfer mechanism of spray cooling is complex, and many factors affect the process's heat transfer characteristics. Grissom and Wierum [4] identified three states in the process of spray cooling—flood mode, dry wall mode, and Leidenfrost mode—with the flood mode being similar to pool boiling. Hsieh and Tien [5] determined the droplet velocity distribution in the spray cooling process and found that the average droplet velocity directly affects heat transfer under nonboiling conditions. Pais et al. [6] conducted experiments on different rough surfaces and proposed that if the roughness of the heat sink surface is less than 1 μm, the main heat transfer mechanism is heat conduction and liquid film evaporation, while that of a rougher surface is mainly nucleate boiling heat transfer. Ortiz and Gonzalez [7] experimented with water as a working medium and found that the heat flux increased with the mass flow and surface roughness, while it decreased with an increase in undercooling and the impact angle. Wang et al. [8] studied water spray experimentally and found that increasing the surface temperature increases the evaporation capacity of the liquid film and improves the heat transfer performance. They also observed that increasing the spray angle gradually improves the heat transfer performance and liquid utilization efficiency. Cheng et al. [9,10] studied the heat transfer in water spray cooling under nonboiling conditions and noted that increasing the spray flow and reducing the spray height can enhance heat transfer. Rybicki and Mudawar [11] conducted single phase spray experiments with PF-5052 and found that increasing the flow or volume flux improves the cooling performance and that reducing the droplet

diameter promotes heat transfer. Abbasi and Kim [12] used PF-5060, PAO-2, and PSF-3 as working fluids to conduct spray cooling experiments with different spray distances, spray pressures, and nozzles and found that the heat transfer in the single phase region is closely related to the impact pressure. Silk et al. [13–15] conducted spray cooling experiments on different surface structures using PF-5060 as the working fluid. They found that enhancing the surface structure increases the heat exchange surface area and provides more nucleation points that enhance heat transfer. They proposed that heat transfer is related not to the wetted surface area but to the utilization efficiency of the wetted area. Bostanci et al. [16,17] performed a spray cooling experiment with a microstructured surface and a smooth surface of ammonia. They found that the heat transfer capacity of the microstructured surface exceeded that of the smooth surface and obtained a critical heat flux (CHF) of 500 $W \cdot cm^{-2}$ and a heat transfer coefficient of 47 $W \cdot cm^{-2} \cdot K^{-1}$ for the microstructured surface. Awonorin [18] measured the rates of heat transfer of the individual droplets of liquid nitrogen falling in at different ambient temperatures. They found that temperature, flow rate, and spray height affect the evaporation of liquid nitrogen droplets and liquid nitrogen evaporates faster at higher spray heights. Tilton et al. [19] conducted liquid nitrogen spray experiments and obtained a CHF of 75 $W \cdot cm^{-2}$. They observed that the heat transfer coefficient of liquid nitrogen spray cooling increases sharply with an increase in the surface temperature. Sehmbey et al. [20,21] examined the heat transfer of liquid nitrogen spray with four different nozzles under different spray conditions. The CHF and heat transfer coefficient increased with the mass flow and with a decrease in the nozzle aperture, and the CHF was found to be 165 $W \cdot m^{-2}$. Somasundaram and Tay [22] performed an experiment in which liquid nitrogen was sprayed intermittently. It was found that with high heat fluxes, the spray height matters significantly, and sufficient height is important in order to cover the whole heat exchange area. Furthermore, surface temperature fluctuations are more pronounced at high heat flux density. She et al. [23,24] studied the relationship between the superheat, heat flux, and heat transfer coefficient of liquid nitrogen spray. An increase in the heat flux caused the increase rate of the surface temperature to gradually increase, and the increase rate was found to be related to the heat exchange area. Furthermore, increasing the flow flux could significantly improve the heat transfer and increasing the heat exchange area reduced the frequency and intensity of droplets impinging per unit area, which weakened the heat exchange effect.

Liquid nitrogen is an easily gasified and highly wetting fluid, and the studies of liquid nitrogen sprays focused on the influence of flow characteristics, spray mass flow rate, surface characteristics, spray pressure and other single parameters. In this study, a bench was constructed for liquid nitrogen spray cooling experiments which were conducted to study the application of spray cooling to the cryogenic field. In particular, the effects of mass flow, spray height, and heat sink surface characteristics on the heat transfer characteristics of liquid nitrogen spray cooling were analyzed. The results of this study are expected to improve existing understanding of the heat transfer mechanism and factors influencing liquid nitrogen spray cooling, apart from serving as a reference for engineering applications of the technique.

## 2. Experimental Setup and Process

An open loop system was used in the experiment, shown in Figure 1. The spray chamber comprised internal and external parts, as shown in Figure 2, and the inner chamber was a cylindrical stainless steel tub. The pedestal and chamber were connected by flanges and insulated with 50 mm polyurethane foam. The outer chamber consisted of two semicylindrical stainless steel tubs connected by a hinge, and it could provide a low temperature space outside the inner chamber to strengthen the cold insulation; it was insulated with a 50 mm rubber insulation material and a plastic material. The nozzle was a TG 6.5 solid cone pressure atomizer with a nozzle orifice of 2.38 mm diameter. The spray height could be adjusted by varying the length of the spray rod, as shown in Figure 2. The heat source simulation device had a high temperature co-fired ceramic heating device, and the device

and a thin film thermocouple were located in the polytetrafluoroethylene (PTFE) base. Moreover, The specifications of the simulated heat sources used in this study are presented in Figure 3. The heat sink surface temperature was directly measured by the thin film thermocouple (seen in Figure 4); compared with a standard thermocouple, the temperature measurement error of the thin film thermocouple was 0.55–1.29%.

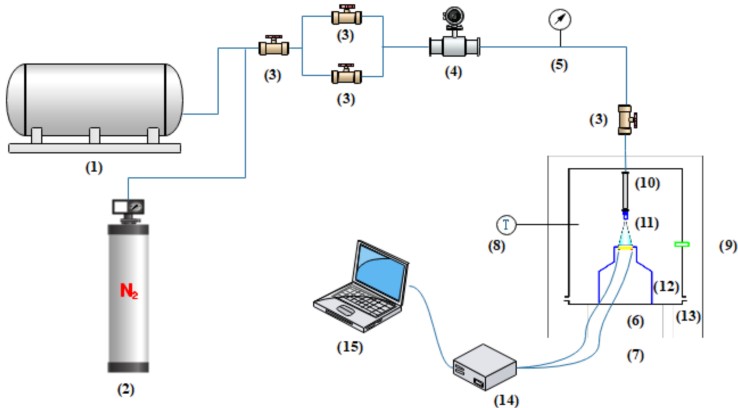

**Figure 1.** Schematic diagram of experimental system.

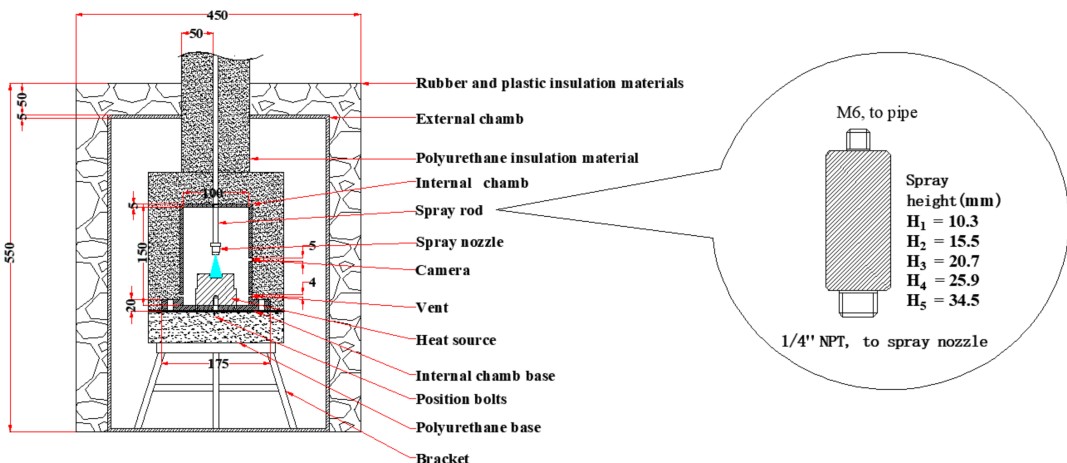

**Figure 2.** Schematic diagram of Spray chamber and Spray rods.

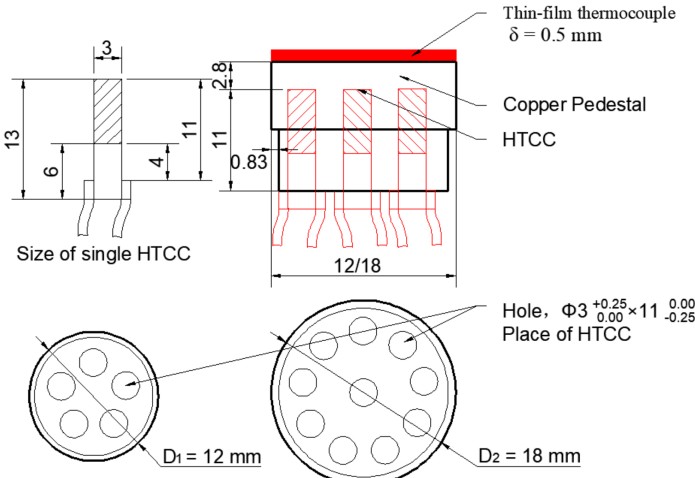

**Figure 3.** Schematic diagram of Heat source simulation device.

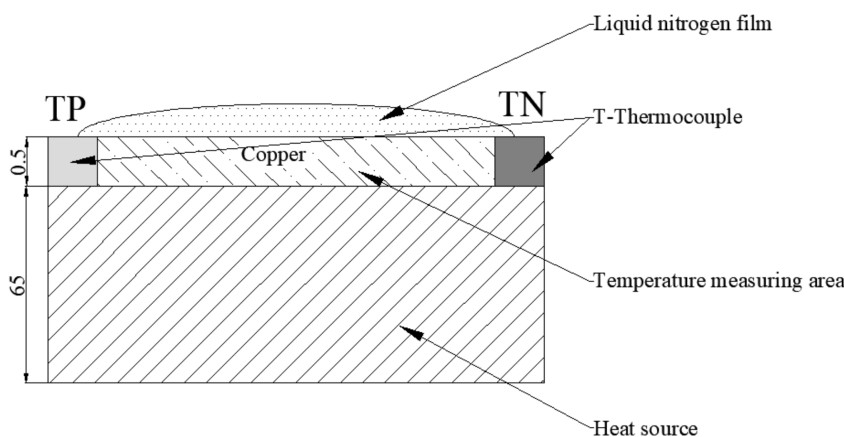

**Figure 4.** Schematic diagram of heat sink surface temperature measurement.

The experiment consisted of the following steps.

(1) After the experimental equipment was connected, the pipeline was purged with nitrogen gas.
(2) The valves shown in Figure 1 were opened, and the liquid nitrogen's flow rate was adjusted. The heating power was then adjusted, and the change in the heat sink surface temperature was observed.
(3) The experiment was stopped when the CHF was reached.
(4) The experimental parameters were changed, and Steps (1)–(3) were repeated.

In the experiments, there are two different orders of magnitude of flow rates, 30 kg·h$^{-1}$ and 15 kg·h$^{-1}$ respectively (hereinafter referred to as high flow rate (HG) and low flow rate (LG)). By considering different heat sink surface diameters (D), spray heights (H), and flow rates, 50 sets of experiments were conducted. The experimental conditions are presented in Table 1.

**Table 1.** The experimental conditions.

| | | $T_{in}$ = 77 K, $T_{envir}$ = 77–78 K, $P_{envir}$ = 101 kPa. | | | | | | | | |
|---|---|---|---|---|---|---|---|---|---|---|
| | | **H (mm)** | | | | | | | | |
| **G (kg·h$^{-1}$)** | | **10.3** | | **15.5** | | **20.7** | | **25.9** | | **34.5** |
| | 12 | 16.00 | 34.01 | 18.56 | 32.18 | 14.76 | 34.36 | 13.51 | 32.11 | 15.85 | 34.90 |
| | 18 | 14.82 | 32.26 | 17.30 | 33.27 | 16.16 | 30.86 | 16.27 | 31.27 | 13.90 | 31.10 |
| **D (mm)** | 24 | 16.09 | 36.03 | 16.71 | 32.78 | 16.71 | 32.78 | 16.09 | 36.03 | 17.95 | 30.95 |
| | 30 | 16.12 | 36.55 | 16.05 | 31.23 | 15.11 | 32.25 | 13.31 | 30.68 | 15.42 | 31.76 |
| | 40 | 17.74 | 33.93 | 15.65 | 33.58 | 16.84 | 33.02 | 17.99 | 33.73 | 15.27 | 32.53 |

## 3. Data Processing and Uncertainty Analysis

The heat transfer coefficient (*h*) was calculated as:

$$h = \frac{q''}{\Delta T_{sat}} = \frac{q''}{T_w - T_{sat}} \tag{1}$$

where $q''$ is the heat flux, $\Delta T_{sat}$ is the superheat, $T_w$ is the temperature of the heat sink, and $T_{sat}$ is the saturation temperature of liquid nitrogen.

On the basis of the law of conservation of energy, the heat flux was calculated using the expression:

$$q'' = \frac{(q_{in} - q_{loss})}{S} = \frac{(UI - q_{loss})}{S} \tag{2}$$

where $q_{in}$ is the input heat flow, $q_{loss}$ is the heat loss of the experiment, $S$ is the area of the heat sink surface, $U$, $I$ are the output voltage and current of the simulated heat source, respectively.

The heat loss was calculated as:

$$q_{loss} = q_{side} + q_{bottom} = \sum A \frac{\lambda_0}{\delta} \Delta T \tag{3}$$

where $q_{side}$ and $q_{bottom}$ are the heat loss of the side wall and bottom of the simulated heat source, respectively, $A$ is the area of the heat loss, $\lambda_0$ is the thermal conductivity of PTFE, which is $0.27$ W·m$^{-1}$·K$^{-1}$, $\delta$ is the thickness of the side/bottom wall of the simulated heat source, $\Delta T$ is the temperature difference between the simulated heat source and the spray chamber.

In the experiment, on the basis of the thickness of the PTFE base (23 mm on the side and 44 mm at the bottom) and the temperature difference between the heat source simulation device and environment, the maximum system heat loss was calculated to be 0.204%.

The heat flux and heat transfer coefficient in the experimental results were determined from the calculated parameters. The accuracy of the T-type thermocouple was $\pm 0.5$ K, and the position accuracy of the thermocouple was determined from the processing technology to be $\pm 0.1$ mm. Furthermore, the uncertainty in the spray distance was $\pm 0.1$ mm, the accuracy of the current and voltage was 0.2%, and the accuracy of the flowmeter was $\pm 0.2$%. The uncertainty was calculated as:

$$R_N = \left\{ \sum_{i=1}^{N} \left( \frac{\delta N}{\delta X_i} R_i \right)^2 \right\}^{\frac{1}{2}} \tag{4}$$

In the experiment, the uncertainties in the heat flux and comprehensive heat transfer coefficient were found to be $\pm 4.48$% and $\pm 5.75$%, respectively.

## 4. Discussion

The spray height and heat sink surface diameter determined the spray impact area ($A_S$) and heat surface area ($A_H$). On the basis of the difference between the two areas, the coverage could be classified into over coverage ($A_H < A_S$), complete coverage ($A_H = A_S$), and incomplete coverage ($A_H > A_S$), as shown in Figure 5.

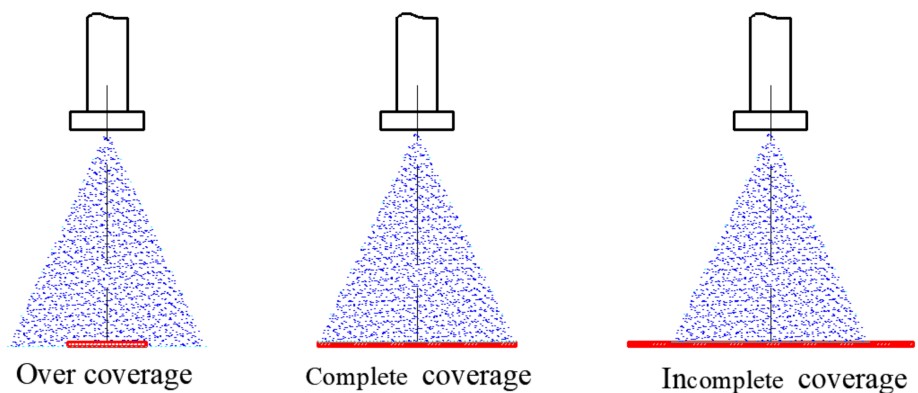

Over coverage  Complete coverage  Incomplete coverage

**Figure 5.** Schematic diagram of Three coverages.

*4.1. Heat Flux, Heat Transfer Coefficient, and Superheat*

4.1.1. Complete Coverage

The change in the heat flux and the heat transfer coefficient with superheat at complete coverage is shown in Figure 6.

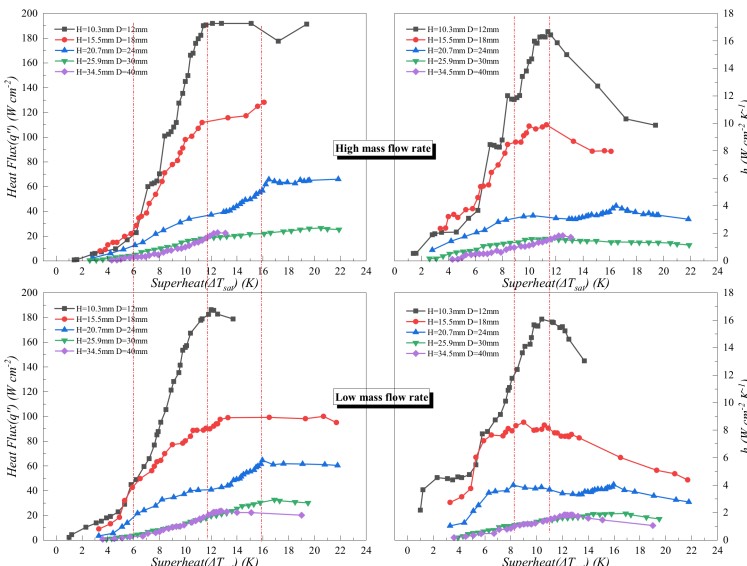

**Figure 6.** Heat flux and heat transfer coefficient versus superheat at complete coverage.

The spray height and heat sink surface diameter of (10.3, 12) corresponded to the maximum heat flux and heat transfer coefficient, and the heat flux and heat transfer coefficient decreased with any further increase in the spray height and heat sink surface diameter. When the spray height and heat sink surface diameter were (25.9, 30) and (34.5, 40), the heat flux and heat transfer coefficient had similar values. For $\Delta T$sat < 6 K, the heat flux increased slowly with an increase in the superheat, indicating a gas–liquid two phase region. In this region, the heat transfer mainly depended on the evaporation and convection of the liquid film. For 6 K < $\Delta T$sat < 12 K, the heat flux increased rapidly with an increase in the superheat and entered the nuclear boiling zone. Here, the heat transfer mechanism was mainly liquid film boiling heat transfer and secondary nucleation. For 12 K < $\Delta T$sat < 16 K, the heat flux was basically stable with an increase in the superheat; it fluctuated only in a small range and entered the critical heat exchange zone close to the CHF. For 12 K < $\Delta T$sat < 16 K, numerous bubbles appeared in the liquid film and a gas film was formed within the liquid film. The spray droplets had to penetrate the gas film to reach the heat sink's surface. After the droplets vaporized, the gas film was supplemented and recovered. The entire heat exchange process was dynamically balanced in the cycle "gas film formation (by evaporation of droplets)–gas film breaking (by the droplets impacting the heat surface)–gas film recovery (by evaporation of droplets)". In the critical heat exchange zone, the relationship between the heat flux and surface superheat was very sensitive. A small increase in the heat flux could destroy the balance and lead to a rapid increase in the superheat. At 8 K < $\Delta T$sat < 11 K, the heat transfer coefficient of spray cooling reached the maximum ($h_{\max}$) in the nucleate boiling region.

In the experiment, the "CHF criterion" [25] was not applicable to liquid nitrogen. For a given heat sink surface diameter, a smaller spray height led to a higher heat flux. This is because the evaporation rate of the liquid nitrogen droplets increased with the increase of the spray height, and the liquid nitrogen flow involved in heat exchange was reduced. The optimal spray height increased with the heat sink surface diameter. When the spray height reached a certain level, the liquid nitrogen droplet gasification rate became too high. Thus, the heat transfer effect at the optimal spray height was not as good as that at a smaller spray height. Mudawar and Estes used water as the working medium in their spray cooling experiment, and its evaporation rate was very low; hence, the CHF criterion could be satisfied.

### 4.1.2. Incomplete Coverage

The change in the heat flux and superheat of the heat transfer coefficient at incomplete coverage is shown in Figure 7.

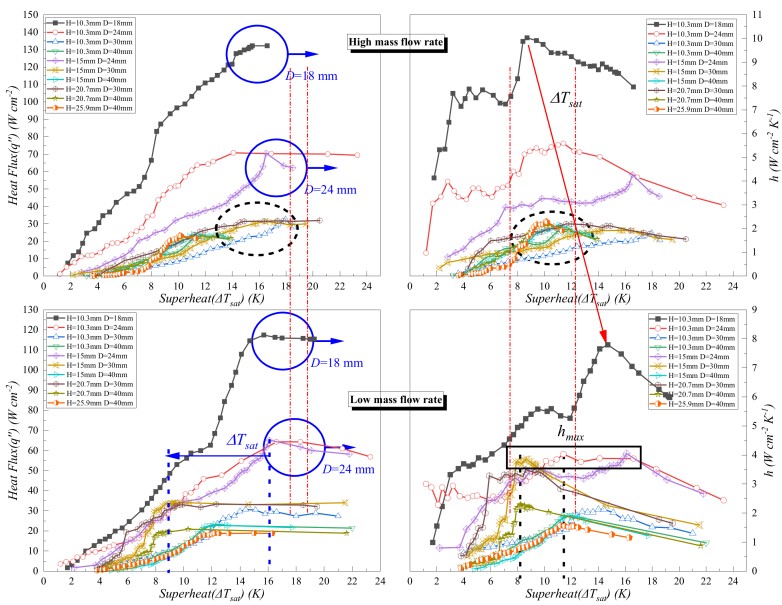

**Figure 7.** Heat flux and heat transfer coefficient versus superheat at incomplete coverage.

At 15 K < $\Delta T$sat < 17 K, all experimental sets entered the CHF region, lagging behind the complete coverage (at 12 K < $\Delta T$sat < 16 K). At 8.5 K < $\Delta T$sat < 15 K, the heat transfer coefficient of all experimental sets reached the maximum. The heat flux and heat transfer coefficient at a high mass flow rate were greater than those at a low mass flow rate, and the superheat of the high mass flow rate at $h_{max}$ was lower than that of the low mass flow rate, indicating that increasing the spray mass flow rate could enhance heat transfer. When the spray height and heat sink surface diameter were (10.3, 18), (10.3, 24), and (15.5, 24), the heat flux and heat transfer coefficient were higher than those of the other seven experimental sets ((10.3, 30), (10.3, 40), (15.5 30), (15.5, 40), (20.7, 30), (20.7, 40), (34.5, 40)). The heat flux and heat transfer coefficient of the other seven experimental sets were similar, as shown in Figure 8. The heat flux and heat transfer coefficient of the seven experimental sets were lower than 35 W·cm$^{-2}$ and 2.5 W·cm$^{-2}$·K$^{-1}$.

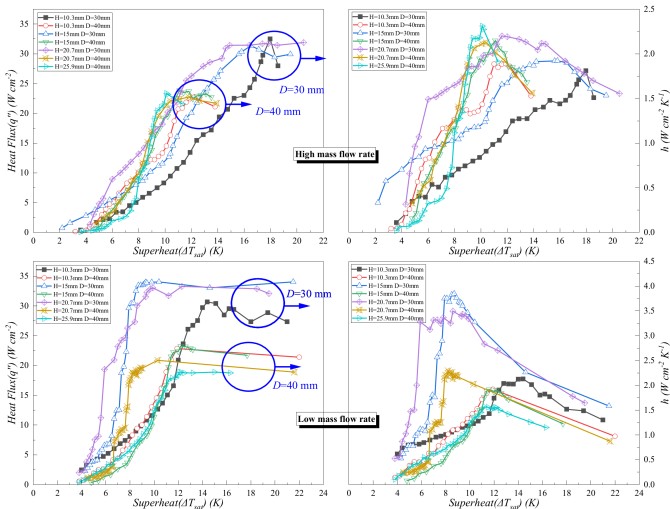

**Figure 8.** Heat flux and heat transfer coefficient versus superheat of seven experimental sets with low heat flux and heat transfer coefficient at incomplete coverage.

At incomplete coverage, the heat sink surface diameter determined the gradient of the heat flux and heat transfer coefficient. The heat flux and heat transfer coefficient could be divided into four levels on the basis of the heat sink surface diameter. The influence of the heat sink surface diameter on the heat transfer was greater than that of the spray height.

### 4.1.3. Over Coverage

The change in the heat flux and superheat of the heat transfer coefficient at over coverage is shown in Figure 9.

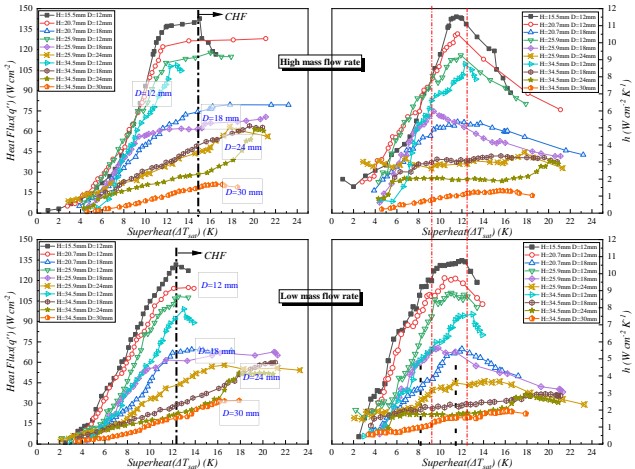

**Figure 9.** Heat flux and heat transfer coefficient versus superheat at over coverage.

At $\Delta T$sat = 15 K, the high mass flow rate experimental sets entered the CHF stage, and at $\Delta T$sat = 12.5 K, the low mass flow rate experimental sets entered the CHF stage. At 9 K < $\Delta T$sat < 12.5 K, the heat transfer coefficient reached the maximum. For high mass flow rate at $\Delta T$sat < 6 K (low mass flow rate at $\Delta T$sat < 5 K), the heat flux increased slowly with an increase in the superheat, and at 6 K < $\Delta T$sat < 12 K (low mass flow rate at 5 K < $\Delta T$sat < 12.5 K), the heat flux increased rapidly with an increase in the superheat. In particular, for different mass flow rates, with an increase in the heat sink surface diameter, the growth rates of the heat flux and superheat gradually decreased. Similar to the case of incomplete coverage, the two parameters, namely spray height and heat sink surface diameter, could be divided into four levels depending on the extent of increase in the heat sink diameter. Therefore, the influence of the heat sink surface diameter on the heat transfer was greater than that of the spray height.

### 4.2. CHF and $h_{max}$

The change trend and distribution of CHF and $h_{max}$ at different spray heights and heat sink surface diameters under different flow rates are shown in Figures 10 and 11, respectively.

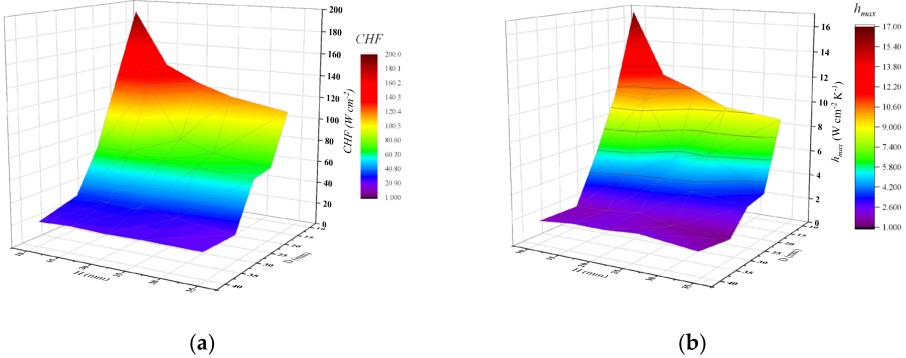

(**a**)                    (**b**)

**Figure 10.** *Cont.*

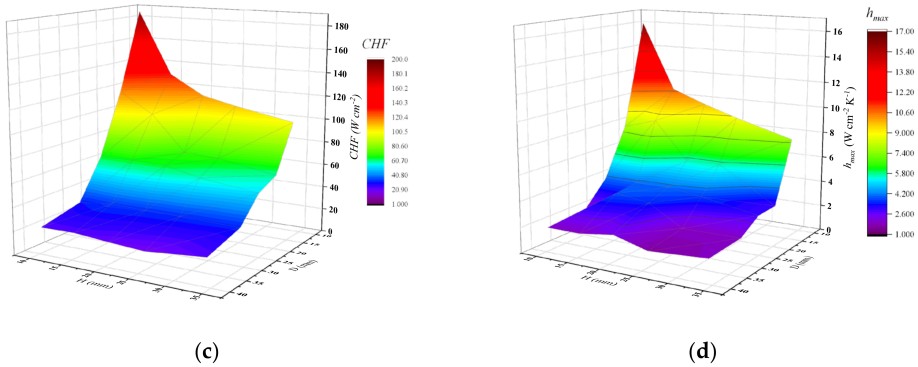

**Figure 10.** Variations of the distributions of CHF and $h_{max}$ at different mass flow rates. (**a**) CHF distribution at high mass flow rate; (**b**) $h_{max}$ distribution at high mass flow rate; (**c**) CHF distribution at low mass flow rate; (**d**) $h_{max}$ distribution at low mass flow rate.

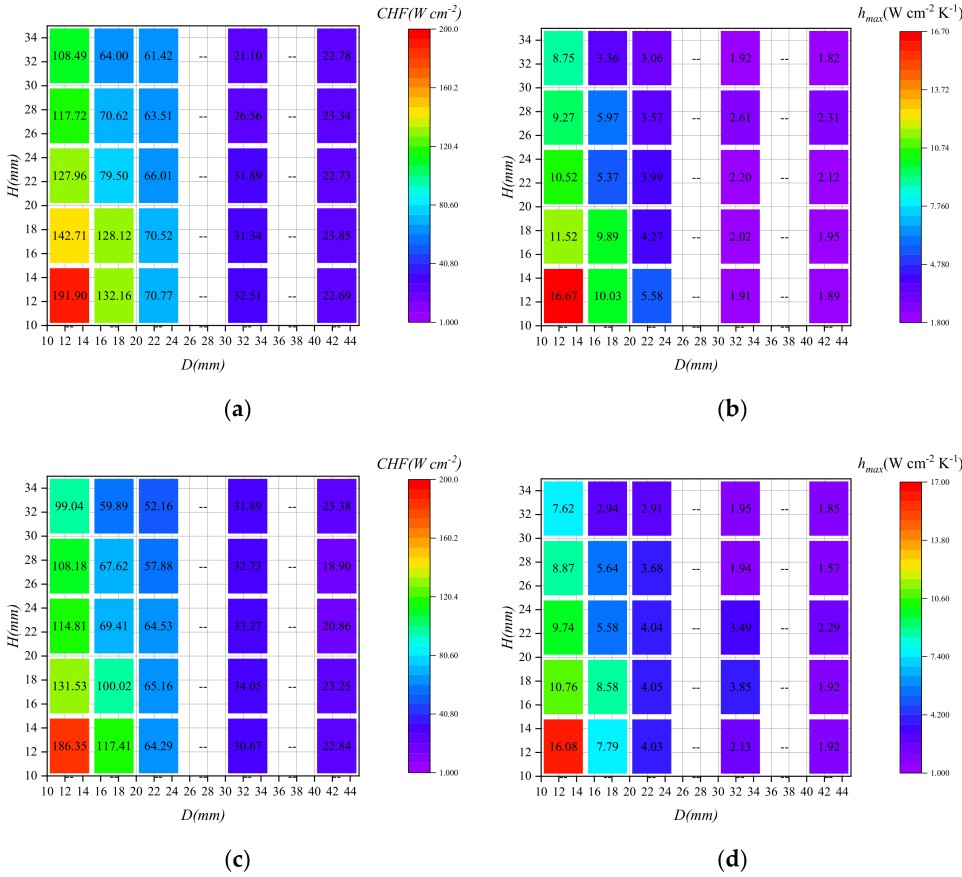

**Figure 11.** Thermodynamic diagrams of the distributions of CHF and $h_{max}$ at different mass flow rates. (**a**) Thermodynamic diagram of CHF distribution at high mass flow rate; (**b**) Thermodynamic diagram of $h_{max}$ distribution at high mass flow rate; (**c**) Thermodynamic diagram of CHF distribution at low mass flow rate; (**d**) Thermodynamic diagram of $h_{max}$ distribution at low mass flow rate.

The CHF and $h_{max}$ decreased rapidly with an increase in the heat sink surface diameter, and slowly with an increase in the spray height. When the heat sink surface diameter was small, the influence of the spray height was apparent. However, when the heat sink surface diameter was large, the influence of the spray height weakened; in particular, when the heat sink surface diameter was greater than 18 mm, the influence of the spray height weakened significantly. In the experiment, for a high mass flow rate of the spray, for $H$ = 10.3 mm and $D$ = 12 mm, the maximum values of CHF and $h_{max}$ were 191.90 W·cm$^{-2}$ and 16.67 W·cm$^{-2}$·K$^{-1}$.

When liquid nitrogen droplet particles were ejected from the nozzle at a high speed, they exchanged heat with the surrounding environment and evaporated, the droplet diameter decreased, and the evaporation speed gradually increased with an increase in the distance of the droplets from the nozzle [18]. This resulted in a reduction in the kinetic energy of the droplets upon contact with the heat sink surface, which limited the spread of the liquid film on the heat sink and reduced the heat exchange capacity. The influence of the heat sink surface diameter on the CHF and $h_{max}$ depended on the amount of working medium droplets falling on the heat sink surface per unit area. For a constant spray mass flow rate, a larger heat sink surface diameter resulted in a smaller flow flux of the working fluid on the surface, which weakened the heat transfer capacity.

The effect of the spray mass flow rate on the heat transfer characteristics was the strongest when the heat sink surface diameter was small. When the heat sink surface diameter was less than 30 mm, for a given spray height, the CHF and $h_{max}$ at a low spray mass flow rate were smaller than those at a high flow rate. The spray mass flow rate also affected the decreasing trend of the CHF and $h_{max}$ when the spray height and heat sink surface diameter increased. The decreasing trend of the two parameters for high flow rate conditions was slightly slower than that for low mass flow rate conditions.

The relationship between the spray impact area and heater surface area is described by the area ratio, which is the spray surface area divided by the heat sink surface area. The distribution of CHF and $h_{max}$ for different spray flow rates and area ratios ($A_s/A_h$) is shown in Figure 12.

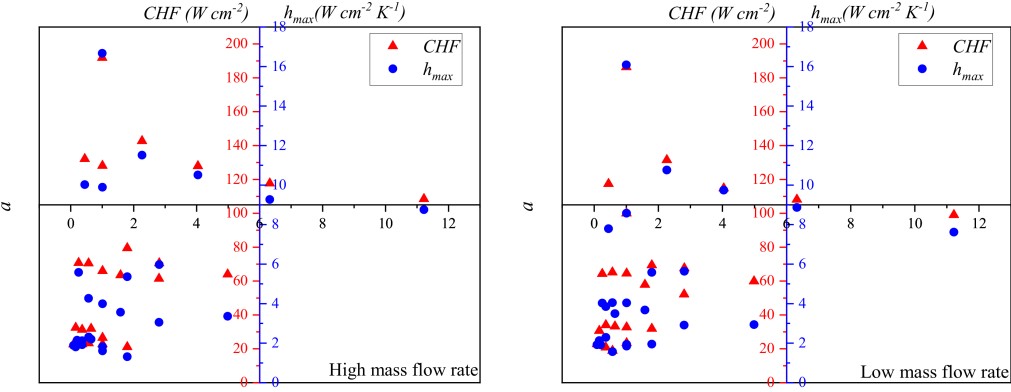

**Figure 12.** The distributions of CHF and $h_{max}$ at different mass flow rates and area ratios ($A_s/A_h$).

When the area ratio is less than one, the spray cannot cover the entire heat sink surface, and hence it is necessary to rely on the spreading and flow of the liquid film on the heat sink for effective heat exchange. Since the CHF and $h_{max}$ are less than 70 W·cm$^{-2}$ and 6 W·cm$^{-2}$·K$^{-1}$, the heat exchange capacity is low. When the area ratio is equal to or greater than one, the CHF and $h_{max}$ are greater than 70 W·cm$^{-2}$ and 6 W·cm$^{-2}$·K$^{-1}$. The distribution of the CHF did not differ significantly for different mass flow rates. There were many points with CHF > 100 W·cm$^{-2}$ and $h_{max}$ > 9 W·cm$^{-2}$·K$^{-1}$ for high flow rate, indicating that increasing the flow rate could improve the CHF and $h_{max}$.

(1) $H$ = 10.3 mm

The evaporation of liquid nitrogen droplets was weak and the droplet diameter was large, and therefore there was sufficient kinetic energy in the droplets to quickly spread on the heat sink and form a liquid film for heat exchange. Furthermore, the thickness and duration of the liquid film were sufficient for the heat sink surface to have satisfactory wetting conditions and promote heat exchange.

$D$ from 12 mm to 40 mm: A larger heat sink surface diameter resulted in a smaller flow flux, which reduced the amount of working medium droplets involved in heat exchange per unit area and hence the heat exchange. For $D$ = 12 mm, the CHF and $h_{max}$ were the

maximum. When $D$ increased from 12 mm to 40 mm, the CHF and $h_{max}$ clearly decreased. For $D$ = 30 mm, the rate of decrease of the CHF and $h_{max}$ decreased, as shown in Figure 13.

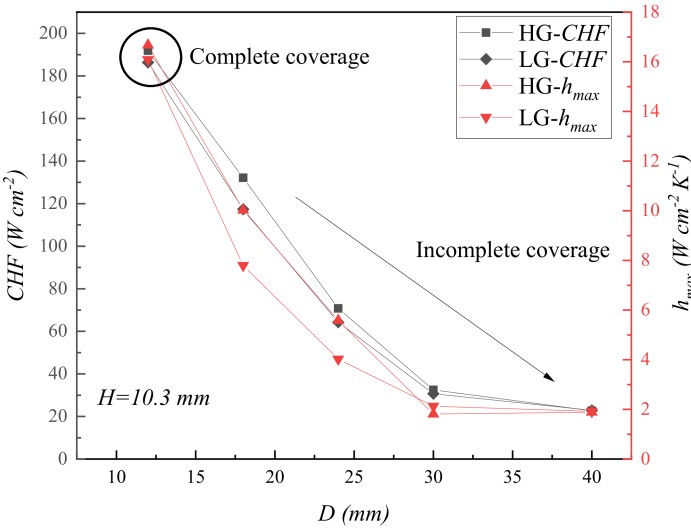

**Figure 13.** *CHF* and *$h_{max}$* versus D at $H$ = 10.3 mm.

(2) $H$ = 15.5–20.7 mm

Increasing the spray height increased the evaporation of liquid nitrogen droplets, which reduced the kinetic energy of the droplets impacting the heat sink; this limited the liquid film's spreading ability and reduced its overall heat transfer ability.

The change in the CHF and $h_{max}$ with the heat sink surface diameter for a spray height of 15.5 mm is shown in Figure 14. For $D$ = 12 and 18 mm, the droplet particles could produce a liquid film covering the entire heat sink surface after impinging on the heat sink surface, and the heat transfer was good. For $D$ = 24 mm, 30 mm, and 40 mm, after the spray droplets impinged on the heat sink surface, a liquid film was formed on the spray impact area, and it spread and covered the entire heat sink surface. A large heat sink surface leads to insufficient spreading of the liquid film on the heat sink surface and low heat exchange capacity. When $D$ increased from 12 mm to 18 mm, the decrease in the CHF and $h_{max}$ was small, mainly because the heat sink surface diameter was small, and the flow flux and heat exchange capacity were high. When $D$ increased from 24 mm to 40 mm, the changes in the CHF and $h_{max}$ were similar to those for $H$ = 10.3 mm.

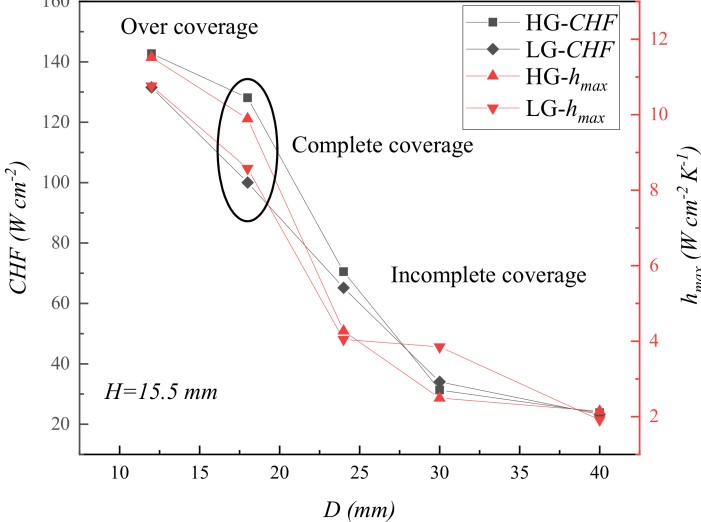

**Figure 14.** *CHF* and *$h_{max}$* versus D at $H$ = 15.5 mm.

The change in the CHF and $h_{\max}$ with the heat sink surface diameter for a spray height of 20.7 mm is shown in Figure 15. The figure clearly shows that the CHF and $h_{\max}$ decreased when the heat sink surface diameter increased from 12 mm to 18 mm. When *D* increased from 18 mm to 24 mm, the CHF and $h_{\max}$ initially showed a slight decrease, and then the CHF decreased rapidly while $h_{\max}$ decreased slowly.

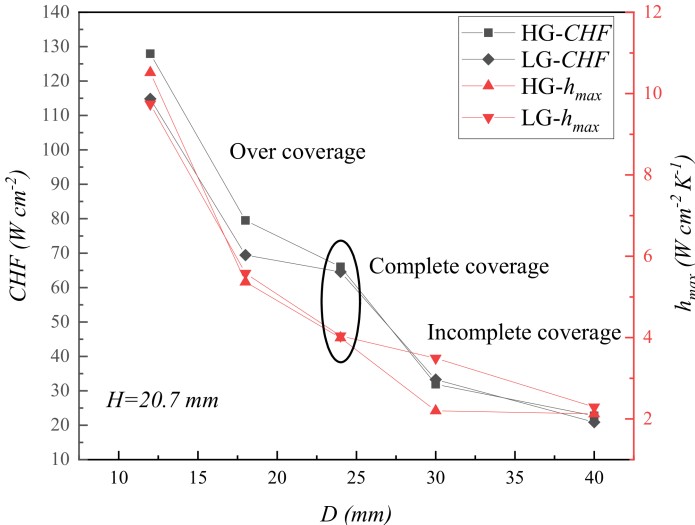

**Figure 15.** *CHF* and $h_{max}$ versus D at *H* = 20.7 mm.

The change in the CHF and $h_{\max}$ with the heat sink surface diameter for a spray height of 25.9 mm is shown in Figure 16. When *D* increased from 12 mm to 30 mm, the CHF and $h_{\max}$ decreased linearly with the increase, and when *D* increased from 30 mm to 40 mm, the CHF and $h_{\max}$ did not change significantly.

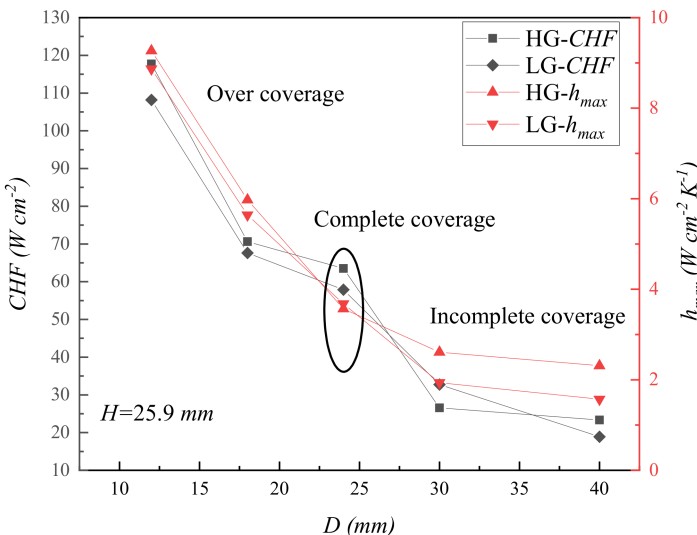

**Figure 16.** *CHF* and $h_{max}$ versus D at *H* = 25.9 mm.

The CHF and $h_{\max}$ in the over coverage area were better than those in the full coverage area, mainly because the edge of the heat sink surface was fully covered, which greatly slowed the local drying process of the heat sink surface in the edge area and maintained the temperature uniformity of the heat sink surface [26]. However, the over coverage resulted in considerable wastage of the working medium; it improved the heat exchange capacity, but increased the cost considerably.

(3) $H$ = 34.5 mm

The change in the CHF and $h_{max}$ with the heat sink surface diameter for a spray height of 34.5 mm is shown in Figure 17. When D increased from 12 mm to 40 mm, the CHF and $h_{max}$ decrease in a stepwise manner.

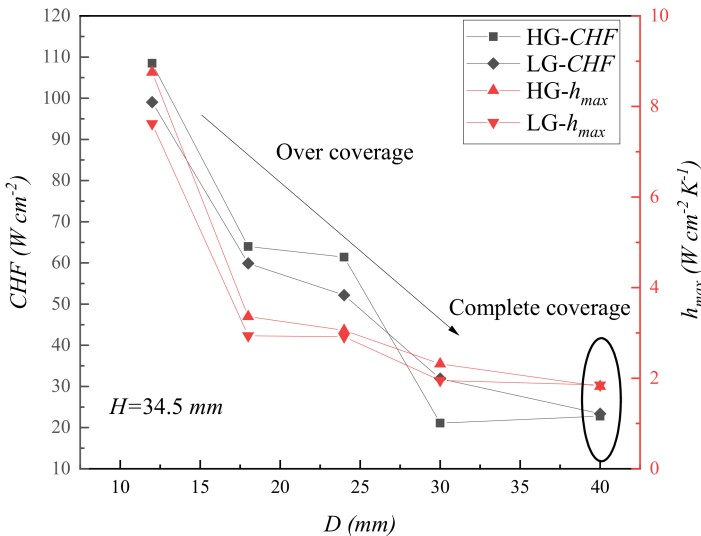

**Figure 17.** *CHF* and $h_{max}$ versus D at $H$ = 34.5 mm.

In general, the heat sink surface diameter determines the spray flow per unit area. The larger the diameter, the smaller the spray flow per unit area, which is detrimental to heat transfer performance. The spray height affects the kinetic energy of the droplet impacting the heat surface. For liquid nitrogen, the higher the spray height, the longer the heat exchange time between the droplet and the surrounding environment before impacting the heat sink surface, and the greater the evaporation of the droplet, so as to reduce the kinetic energy of the droplet. Hence, the spread and flow of the liquid film are reduced, and the heat transfer performance weakens. The coupling of spray height and heat sink diameter affects the ratio of spray surface to heat sink surface. In the combined effect of the heat sink surface diameter and spray height on the heat transfer performance of spray cooling, the heat sink surface diameter plays a more dominant role. The effect of the spray flow rate on the heat transfer characteristics decreased with an increase in the spray height.

### 4.3. Correlations

Finally, in order to make it easier to use the results in this article for practical or academic purposes, they are also offered in mathematical form. The surface superheat, the mass flow rate, spray height, and the heat sink surface diameter were all mathematically correlated. The correlation was as follows:

$$\mathrm{Nu} = 6.2110\mathrm{Re}^{0.7964}\mathrm{We}^{-0.0965}\mathrm{Bg}^{0.5898} \tag{5}$$

where Nu, Re, We, and Bg are Nusselt number, Reynolds number, Weber number, and boiling number, respectively, expressed as:

$$\mathrm{Nu} = \frac{hD}{\lambda} = \frac{q''}{\Delta T_{sat}}\frac{D}{\lambda} \tag{6}$$

$$\mathrm{Re} = \frac{G''D}{\mu} \tag{7}$$

$$\mathrm{We} = \frac{\rho u_0{}^2 d_{32}}{\sigma} \tag{8}$$

$$\mathrm{Bg} = \frac{q''\,S}{h_{lg}G} \tag{9}$$

The spheres of application of the correlation are as follows: Re = 800–6000, We = 1200–20,000 and Bg = 0.0005–0.35. The correlations predict values presented in this article within 20.77%.

## 5. Conclusions

In this study, liquid nitrogen was used as the spray working medium in a series of experiments, and the combined effect of the spray height and heat sink surface diameter on the cooling performance was focused upon. The main conclusions of this study are as follows.

The heat sink surface diameter affected the spray flow per unit area. A larger heat sink surface diameter resulted in a smaller spray flow per unit area, which was unfavorable for heat exchange. Furthermore, the spray height affected the droplet diameter; a larger spray height resulted in a longer heat exchange between the droplet particles and the surrounding environment before the droplets impinged on the heat sink surface and also increased droplet evaporation. This reduced the kinetic energy of the droplets impinging on the heat sink surface, which reduced the spreading and flow of the liquid film and eventually weakened the heat exchange capacity of the liquid film.

The combined effect of the spray height and heat sink surface diameter was found to affect the ratio of the spray surface area to the heat sink surface area. On the basis of the difference between the spray height and heat sink surface diameter, the spray could be categorized into three types: over coverage, complete coverage, and incomplete coverage. The heat flux decreased in a stepwise manner with a direct increase in the heat sink surface. For a given heat sink surface diameter, it decreased with an increase in the spray height. Furthermore, a smaller heat sink surface diameter led to a more apparent change in the spray height. In general, the influence of the heat sink surface diameter on heat transfer was greater than that of the spray height. An increase in the spray flow rate was found to improve the CHF and $h_{\max}$. When approaching the CHF, we can increase the spray flow rate to reduce the surface superheat and delay the CHF.

In the experiment, the max CHF and heat transfer coefficient were 191.90 W·cm$^{-2}$ and 16.67 W·cm$^{-2}$·K$^{-1}$ at the spray height of 10.3 mm and the heat sink diameter of 12 mm.

The droplet evaporation mechanism and the impact droplet lifetime will be investigated in future by visualization experiment of liquid nitrogen spray cooling. The best range of spray height and heat sink surface diameter will also be explored in greater depth at the same time.

**Author Contributions:** Y.S.: Methodology, software, writing—original draft preparation, writing—review & editing, data curation. Y.J.: conceptualization, visualization, investigation, funding acquisition, resources. All authors have read and agreed to the published version of the manuscript.

**Funding:** This research received no external funding.

**Data Availability Statement:** Not applicable.

**Acknowledgments:** We are very grateful to the editors for their careful editing and for providing us with their valuable comments that helped us improve the quality of the manuscript.

**Conflicts of Interest:** The authors declare no conflict of interest.

## Abbreviations

Nomenclature

| | |
|---|---|
| $A$ | the area of side and bottom (m) |
| $Bg$ | boiling number |
| $c_p$ | specific heat at constant pressure (j·kg$^{-1}$·K$^{-1}$) |
| $d_{32}$ | Sauter mean diameter (m) |
| $D$ | surface diameter (m) |

| | |
|---|---|
| $Ds$ | diameter of Spray bottom diameter (m) |
| $G$ | mass flow rate ($kg \cdot s^{-1}$) |
| $G''$ | mass flow flux ($kg \cdot s^{-1} \cdot m^{-2}$) |
| $h$ | heat transfer coefficient ($W \cdot m^{-2} \cdot K^{-1}$) |
| $h_{lg}$ | Latent heat of vaporization ($J \cdot kg^{-1}$) |
| $H$ | nozzle to surface distance (m) |
| $I$ | current (A) |
| $N$ | measured value |
| $Nu$ | Nusselt number |
| $P$ | pressure (Pa) |
| $Q$ | volumetric flow rate ($m^3 \cdot s^{-1} \cdot m^{-2}$) |
| $q''$ | surface heat flux ($W \cdot m^{-2}$) |
| $Ra$ | surface roughness ($\mu$m) |
| $R$ | uncertainty |
| $Re$ | Reynolds number |
| $S$ | area of heat sink surface ($m^2$) |
| $T$ | Temperature (K) |
| $\Delta T$ | $T_{aver}$-$T_w$ (K) |
| $\Delta T_{sat}$ | surface superheat, $T_w$-$T_{sat}$ (K) |
| $u_0$ | droplet velocity ($m \cdot s^{-1}$) |
| $U$ | voltage (V) |
| $We$ | Weber number |
| $X$ | average value |
| *Greek* symbols | |
| $\delta$ | the thickness of PTFE (m) |
| $\lambda$ | thermal conductivity of liquid ($W \cdot m^{-1} \cdot K^{-1}$) |
| $\mu$ | viscosity of liquid ($N \cdot s \cdot m^{-2}$) |
| $\lambda_0$ | thermal conductivity of PTFE ($W \cdot m^{-1} \cdot K^{-1}$) |
| $\sigma$ | Surface tension of liquid ($N \cdot m^{-1}$) |
| Subscripts | |
| *aver* | average |
| *bottom* | bottom surface |
| *envir* | environment |
| *H* | heat surface |
| *in* | input |
| *l* | liquid |
| *loss* | loss |
| *s* | spray |
| *sat* | saturation |
| *side* | side surface |
| *sub* | subcooling |
| *w* | surface of heat sink |

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
