# Peer review of "Liquid-Nitrogen Spray-Cooling Combined Effect of the Spray Height and Heat-Sink-Surface Diameter"

_applsci, doi:10.3390/app12146941_

Round 1

Reviewer 1 Report

The paper discusses liquid nitrogen (LN2) spray cooling for electronic equipment and evaluates the impact of the spray height, heat sink surface diameter, and spray mass flow on the heat transfer characteristics of spray cooling. This work is interesting. However, there are some questions about the assumptions and calculations, so it could be publishable after revising all the problems clearly. 

  1. The literature review is not significantly updated, focusing on LN2 spray cooling technology. Authors must add additional recent studies on LN2 spray cooling in the Introduction section. Discussion on terms like 'LN2 droplet vaporization', continuous/ intermittent spray, etc., may add value to the manuscript.
  2. The motivation for this work, and its novelty, must be adequately stated. 
  3. Which electronic device does LN2 spray cooling find an application for? What is the desired operating temperature of such electronic equipment? Why is the cryogenic temperature required?
  4. What is the pressure inside the chamber? How does spray cooling influence the pressure of the chamber? How is pressure inside the chamber maintained?  
  5. From a reader's point of view, reading the legends in graphs is very difficult.
  6. Figure captions need improvement.  
  7. Fig. 7 shows that when the spray height and heat sink surface diameter are at their respective lowest values (10.3, 12), maximum heat flux and heat transfer coefficient are obtained. Why the impact of further lower values of spray height (below 10.3) and heat sink surface diameter (below 12) has not been studied? Is there any practical limitation to going below (10.3, 12)? Also, it will be good that if the range/limits of values selected in Table 1 are justified.  
  8. Another drawback I felt is that the manuscript is silent on droplet vaporization mechanism and the impact droplet lifetime.
  9. Conclusion section: Scope of future work and some recommendations regarding LN2 spray cooling technology may add value to the manuscript. 

Reviewer 2 Report

  • The paper deals with an interesting topic.

However, some comments need to be addressed :

  1. Please give a thorough review of the English language, all grammatical errors should be carefully eliminated.
  2. The Word format of the manuscript must be revised and performed (example: spaces between words and punctuation in general...)
  3. The original points must be explained well and Highlighted.
  4. Nomenclature and Abreviation list must be added.
  5. Check title of Figure 2 in page 3
  6. Figure 3 is repeated in Figure 2! Please verify it!
  7. Check table1, are the values given in this table of G or D?
  8. Check sentence 132-133,  verify syntax!
  9. Check relation 4. What is the meaning of   dN/dx  ?  
  10. Check line 181 in page 6, justify why “CHF criterion” is not applicable to liquid nitrogen?
  11. Check Fig 12 b: why values of hmax decrease from 16,67 to 1,81 and increase from 1,81 to 1,89 for H=12 to 14m?
  12. Same note for H=26 to 30 or H=30 to 34?
  13. Check Figures 15 and 18, verify curve of HG-CHF and specially value of CHF at D=30mm?  
  14. There is no indication about “measure method” used in this work, can you give more details? Can you give more details about “measurement precision” in the experimental part?
  15. Please, can you give more details about effect of speed magnitude of spray and specially in case of incomplete coverage? What about recirculation zone at high speed?
  16.  Check page 14 (Line 333-338), can you more details about phase change phenomena in your application?
  17. The conclusion section can be performed and give practical recommendations!

Reviewer 3 Report

The paper describes in a very detailed way the series of experiments done to study the heat  transfer in a cooling process done through nitrogen spray. The level of detailed explanation of the work done is adequate and figures (specially 1 to 5) help to understand clearly the process.

The paper is interesting and deserves publication. There are several points that the authors should clarify before doing it

1.- In the first page of the paper appears that the journal the paper is submitted  to is “Applied Sciences”. However, from page 2 onwards the title “Processes” appears in the top pf each page.

2.- Line 63: typo units should be W cm-2 I guess

3.- Table 1: Please, specify what the results are in the table (CHF? Convection coefficient?) and their units

4.- Line 131:  0.27 is repeated

5.- Lines 339 to 342 seem to be out of context. They look like the advice of a previous reviewer (from Processes?) to the present work

6.- The paper will improve a lot if analytical expressions or formulae to calculate h and CHF as a function of the Height, mass flow , area ratio… obtained from the experimental results would be available.

Round 2

Reviewer 1 Report

The authors addressed my questions. It can be accepted.